# Prediction of Epidemic Peak and Infected Cases for COVID-19 Disease in Malaysia, 2020

**DOI:** 10.3390/ijerph17114076

**Published:** 2020-06-08

**Authors:** Abdallah Alsayed, Hayder Sadir, Raja Kamil, Hasan Sari

**Affiliations:** 1Department of Electrical and Electronic Engineering, Faculty of Engineering, Universiti Putra Malaysia, Serdang 43400, Selangor, Malaysia; 2Department of Computer and Wireless Communication, Faculty of Engineering, Universiti Putra Malaysia, Serdang 43400, Selangor, Malaysia; haydersader@gmail.com; 3Laboratory of Computational Statistics and Operations Research, Institute for Mathematical Research, Universiti Putra Malaysia, Serdang 43400, Selangor, Malaysia; 4College of Computer Science and Information Technology, Universiti Tenaga Nasional, Kajang 43000, Malaysia; hassan_sari@yahoo.com

**Keywords:** COVID-19, SEIR model, epidemic peak, infection rate, basic reproductive number, ANFIS, GA

## Abstract

The coronavirus COVID-19 has recently started to spread rapidly in Malaysia. The number of total infected cases has increased to 3662 on 05 April 2020, leading to the country being placed under lockdown. As the main public concern is whether the current situation will continue for the next few months, this study aims to predict the epidemic peak using the Susceptible–Exposed–Infectious–Recovered (SEIR) model, with incorporation of the mortality cases. The infection rate was estimated using the Genetic Algorithm (GA), while the Adaptive Neuro-Fuzzy Inference System (ANFIS) model was used to provide short-time forecasting of the number of infected cases. The results show that the estimated infection rate is 0.228 ± 0.013, while the basic reproductive number is 2.28 ± 0.13. The epidemic peak of COVID-19 in Malaysia could be reached on 26 July 2020, with an uncertain period of 30 days (12 July–11 August). Possible interventions by the government to reduce the infection rate by 25% over two or three months would delay the epidemic peak by 30 and 46 days, respectively. The forecasting results using the ANFIS model show a low Normalized Root Mean Square Error (NRMSE) of 0.041; a low Mean Absolute Percentage Error (MAPE) of 2.45%; and a high coefficient of determination (R^2^) of 0.9964. The results also show that an intervention has a great effect on delaying the epidemic peak and a longer intervention period would reduce the epidemic size at the peak. The study provides important information for public health providers and the government to control the COVID-19 epidemic.

## 1. Introduction

Coronavirus disease (COVID-19) is an infectious disease first reported in China [1]. COVID-19 has been confirmed on 25 January 2020 in Malaysia and currently continues to spread fast in the country, which seriously jeopardizes the lives of elderly people as well as those of any age who experience a serious underlying medical condition [2]. Figure 1 shows the accumulated number of infected cases due to COVID-19 from 25 January to 05 April in Malaysia. It can be observed that the COVID-19 outbreak started to be a pandemic after 27 February, such that more than 98.77% of the total infected cases was reported after this date. This outbreak is mostly attributed to a special religious gathering of more than 15,000 persons between 27 February and 2 March at a local mosque, which was an infection cluster and the main source of the spike in COVID-19 cases according to the Ministry of Health in Malaysia [3]. The spread of the virus came from the foreign participants who came into Malaysia and participated in the gathering. The sudden increase in the number of infected cases after 12 March is probably due to the fact that infected people without COVID-19 symptoms could significantly spread the infection [4]. Furthermore, the diagnostic tests were initially only made available to those who attended the religious gathering.

Unfortunately, COVID-19 cannot be controlled as there is no proven pharmaceutical-based treatment up to now. However, other behavioral strategies, such as lockdown and movement control of people, can be effective to reduce the number of new cases and delay the epidemic peak. The Malaysian Government has promulgated the restricted activities order on 18 March, which prohibits all mass movements and gatherings across the country, including religious, sports, social, and educational activities. The movement control order was implemented in several stages with the strictness and punishment increasing with each stage to ensure that the public conform to the restrictions. However, exclusions are given to public markets, grocery stores, and convenience stores selling food and essential items. The main public concern is whether the epidemic will continue until August 2020, which would affect the economy, and in particular the tourism plan of “Visit Malaysia 2020” that attracts Middle Eastern and Chinese tourists during the holiday season from June through August. Therefore, the short- and long-term prediction of the COVID-19 epidemic is needed to provide important information for healthcare providers and government that would help them to implement effective intervention measures and policies.

Mathematical modeling plays an important role in predicting the epidemic peaks of COVID-19 using real-time historical data [5]. Many statistical and numerical models have been used to predict the COVID-19 outbreaks, such as the Logistic Growth model [6], stochastic Susceptible–Infectious–Removed (SIR) model [7], and Natural Growth model [8]. However, the SEIR (Susceptible–Exposed–Infectious–Recovered) model is still the most widely used to characterize the epidemic peak of COVID-19 in China [9,10,11], Japan [12], Italy [13], and Iran [14]. Besides, the SEIR model was used to compare the effect of the lockdown of Hubei province on the infection rates in Beijing and Wuhan [15]. On the other hand, in forecasting the number of infected cases for the upcoming few days, the mathematical models are not effective as many parameters should be daily updated and estimated. Thus, the accuracy of short-time forecasting using parametric models may not be high [16].

The infection rate (or transmission rate) parameter provides information on the probability of transmission of COVID-19 from an infectious individual to susceptible individuals [17]. It is one of the two components in the basic reproductive number by which the continuous increase or decrease in the infected cases is decided. In calculating the infection rate, the most common method is the asymptotic statistical theory [18], in which the least-squares method is used to quantify the uncertainty associated with infection rate estimation. However, the least-squares method is subjected to low accuracy that accompanies the estimation of the infection rate. A possible solution is to run the estimation process 10,000 times and then obtain the normal distribution of the infection rate values with 95% confidence intervals, which would decrease the uncertainty and increase the accuracy as in [12,19]. This method highly increases the time of the estimation process, especially when the range of the hypothesized infection rate is relatively large with a small resolution.

To our knowledge, there are no scientific studies related to the pandemic of COVID-19 spread in Malaysia. Thus, this study is conducted to (1) estimate the infection rate using the Genetic Algorithm (GA); (2) predict the epidemic peak of COVID-19 using the SEIR model, incorporating also the mortality in the population due to COVID-19; and (3) forecast the number of infected cases for the upcoming five days using the Adaptive Neuro-Fuzzy Inference System (ANFIS) predictive model. The available data of infected cases from 25 January to 05 April 2020 in Malaysia was used to calibrate the SEIR model. For forecasting, the data from 22 to 31 March was used to train and test the ANFIS model, while the data from 01 to 05 April was used to validate the ANFIS model.

## 2. Methods

### 2.1. SEIR Model for Peak Prediction

The SEIR model that characterizes the epidemic COVID-19 outbreaks is described as follows [20,21]:*dS(t)/d(t) = −βS(t)I(t),**dE(t)/d(t) = βS(t)I(t) − αE(t),**dI(t)/d(t) = αE(t) − γI(t) − MI(t),**dR(t)/d(t) = γI(t),**dD(t)/d(t) = MI(t)*(1)
where *S*, *E*, *I, R,* and *D* represent the number of susceptible, exposed (not yet infectious), infective, recovered, and death cases given at time *t* > 0. The coefficients *β*, *α*, *γ*, and *M* denote the infection, onset, removal, and mortality rates. Based on the recent studies related to COVID-19 [22,23,24,25], the incubation (*α*^−-1^) and infectious (*γ*^−1^) periods are 5 days and 10 days, respectively. Thus, the *α* and *γ* values are 0.2 and 0.1, respectively. The total number of deaths and confirmed cases up to 5 April are 61 and 3662, respectively, and thus the mortality rate *M* is 0.016 (61/3662). We fixed the unit time to be 1 day and S + *E* + *I + R + D* = 1, such that each population implies the proportion to the total population. Let assume that there is one infected case recorded at time *t* = 0 among the Malaysian population of *N* = 32.6 × 10^6^ [26]; that is, *X(0) = pNI(0)* = 1, where
*X(t) = pNI(t),*(2)
where *X* is the number of infected cases that are identified at time *t*, and *p* is the identification rate such that we obtain *I*(0) = 1/(*p* × 32.6 × 10^6^). The block diagram of the SEIR model is attached in Appendix A as Figure A1. It is assumed that there are no exposed, recovered, and death cases at *t* = 0, and hence,
(3)$$S(0)\;=\;1-(E(0)+I(0)+R(0)+D(0))\;=\;1-\;\frac{1}{pN}$$

In Malaysia, the COVID-19 test is mainly performed on those with close contacts to the patients as well as on those with COVID-19 symptoms. We assume that the identification rate is not significantly dependent on the test kit availability as the Malaysian government is able to perform the test for 11,500 persons a day and the average number of daily tested cases is 2500 persons [27]. In [28], 3662 cases are currently confirmed among the tested 43,595 infected cases from 25 January to 05 April. Based on that, *p* is equal to 0.084 (3662/43,595). The basic reproductive number *ℛ*_0_ represents the expected number of secondary cases resulted from an infected individual [29]. It is calculated as the leading eigenvalue of the next generation matrix *G* = *FV*^−1^ [30], where
(4)$$F=\left[\begin{array}{cc} 0 & \beta S\left(0\right)\\ 0 & 0 \end{array}\right],\;V=\;\left[\begin{array}{cc} \alpha & 0\\ -\alpha & \gamma \end{array}\right],$$
where *F* is a new infection, while *V* represents the transfers of infections from one compartment to another [5]. Then, we obtain
(5)$${ℛ}_{0}\;=\;\frac{\beta S(0)}{\gamma }\;=\;\frac{\beta }{\gamma }(1\;-\;\frac{1}{pN})\;\approx \;\frac{\beta }{\gamma }.$$

It is obvious that the basic reproductive number only depends on the infection rate (*β*) and the removal rate (*γ*). Besides, the influence of the identification rate on ℛ_0_ is negligible as the population number (*N*) is 32.2 × 10^6^. The coefficient parameters of the SEIR model are summarized in Table 1. Note that the estimation of *β* and ℛ_0_ values are presented in the next subsection.

### 2.2. β. Estimation Using GA

In this study, the estimation of the infection rate is accomplished using the Genetic Algorithm [19]. Let us assume *X(t)* (described in Equation (2)), *t* = 0, 1, …, 72, is the number of daily infected people due to COVID-19 in Malaysia from 25 January to 05 April. We assume *X(t)* is subjected to the Poisson noise, which reflects the fluctuations of the number of infected cases, so that
(6)$$\dot{X}\left(t\right)\;=\;X(t)\;+\;\varepsilon X{\left(t\right)}^{\xi },\;\mathrm{Poisson}\;\mathrm{noise}\;=\;\varepsilon X{\left(t\right)}^{\xi },$$
where $\dot{X}$ is our deterministic model with Poisson noise, while *ε* is a random variable from a normal distribution with a mean zero and a standard deviation of 1. The *ξ* is equal to 0.5, such that the variance of the error scales is linear with *X(t)* and this value refers to the Poisson noise as described by [31]. The classical GA was applied to estimate the *β* value that minimizes the cost function. The cost function is represented by the sum of squares, as in Equation (7). The *β* value ranged from the lower bound to upper bound values. The lower and upper bounds of the *β* value were selected as 0.2 and 0.4, respectively. The minimum cost function *C(β**^″^**)* is defined as in Equation (8).
(7)$$C(\beta )\;=\;\sum_{t=0}^{72}{\left[X\left(t\right)-\dot{X}\left(t\right)\right]}^{2},$$
*C*(*β*″ ) = *min_0.2 ≤ β ≤ 0.4_ C*(*β*),
(8)

The classical GA algorithm was then implemented to find the optimum *β* values that minimize the cost function using five steps, as follows [32,33,34]:Population initialization: In order to find a solution to the problem of the cost function, the GA initially creates a number of populations that randomly encodes the chromosomes (individuals). Then, the cost values of the generated population are evaluated.Selection: In this process, each individual identified by its associated cost is ranked and the corresponding individual fitness is selected. According to fitness, the best chromosomes from the population are then selected such that better fitness has a bigger chance to be selected. Subsequently, the solutions selected from one population are implemented to form a new population. This process is motivated by the new population potentially being better than the previous one. The selection process is performed using a certain function that fixes the generation gap. The selected individuals are then recombined.Crossover: To make new offspring (children) for the following iteration, the selected individuals (parents) have to undergo a crossover with a crossover probability. However, if there is no crossover performed, the offspring is an exact copy of the parents.Mutation: In this process, the information in the chromosomes is randomly modified. The genes occasionally mutate to be converted to novel genes. Based on mutation, it is possible to control the multifariousness of the population as well as to enhance the search capacity of the search scheme.Evaluation: For each individual, the cost function of the optimization problem is calculated. The stopping criterion of the GA is the number of iterations after which the process is stopped. For each iteration, the *β* value that has the minimum cost function is recorded. The distribution of the *β* values is then approximated by a normal distribution with a mean and standard deviation.

The flowchart of the GA for *β* estimation is demonstrated in Figure 2. The GA parameters are provided in Table 2 and obtained based on the trial and error method. The Optimization Toolbox of the MATLAB^®^ software (MathWorks Inc.) was used to implement and run the GA algorithm.

### 2.3. ANFIS for Short-Term Forecasting

ANFIS is a nonparametric model used to solve a nonlinear problem with a small dataset in one framework. It has a powerful hybrid learning capability using an Artificial Neural Network (ANN) and a Fuzzy Logic model to generate an effective processing tool for prediction [35]. The core element of ANFIS is the Fuzzy Interference System (FIS) that is embedded into a framework of adaptive networks that use “IF–THEN” rules to model the behavior of an uncertain system. These adaptive networks contain a number of adaptive nodes connected through directional links. Each adaptive node has a modifiable parameter updated using the fuzzy learning rule aiming to minimize the errors. In this study, the FIS system uses one input *x* and one output *y*. The ANFIS model structure is shown in Figure 3. The first order Sugeno fuzzy model with fuzzy “IF–THEN” rules is employed as follows [36]:Rule 1: if *x* is *A*_1_ then *y*_1_= *P*_1_*x + r*_1_,(9)
Rule 2: if *x* is *A*_2_ then *y*_2_ = *P*_2_*x + r*_2_.(10)

Layer 1 contains the member functions (MFs) of the inputs and generates the input variables for Layer 2. Each node in this layer is adaptive using Equation (11). The MF type used in this study is the Gaussian function, for which 0 and 1 are the lowest and highest values, respectively.
*Q_i_* = *μ_Ai_* (*x*), where *μ* (*x*) is MF.(11)

Layer 2 is a membership layer in which the weights of MFs are computed and considered. Input variables of this layer are obtained from the first layer. Noted that, the layer’s nodes are fixed nodes. The output of the second layer is a product of all incoming inputs and described as in Equation (12), where *w_i_* represents the weight strength of one rule.
*w_i_* = *μ* (*x*)*_i_ μ* (*x*)*_i+_*_1_ and *i* = 1,2.(12)

In Layer 3 (rule layer), the weight function is normalized and the outputs of this layer are called normalized weights or firing strengths. The normalization is described as:(13)$${\bar{w}}_{i}=\frac{w_{i}}{w_{1}+w_{2}},\;\mathrm{and}\;i=1,2.$$

Layer 4 is the defuzzification layer such that the output from Layer 3 is multiplied with the Sugeno fuzzy rule function as follows:(14)$$Q_{i}^{4}=w_{i}^{*}y=w_{i}^{*}(p_{i}x+r_{i}),$$

Layer 5 is the output layer in which the inputs and outputs from the previous layer are formulated. Furthermore, this layer converts the results into a crisp output. Thus, all incoming inputs are sum up producing the overall output as follows:(15)$$Q_{i}^{5}=\sum_{i}w_{i}^{*}y_{i}=\frac{\sum_{i}w_{i\;}y}{\sum_{i}w_{i}}$$

Noted that the ANFIS MFs parameters are adjusted (tuned) using the hybrid method of backpropagation and least square techniques [37]. The Neuro-Fuzzy Designer of Matlab^®^ Software (MathWorks Inc.) is used to implement the ANFIS parameters that are summarized in Table 3. In this study, as the number of infected cases is nonlinearly changed from day to day, the ANFIS model is used. The ANFIS model forecasts the numbers of infected cases for the upcoming 5 days based on the numbers of infected cases for the last 10 days. The dataset of 10 days is divided into training (70%) and testing (30%) datasets which are implemented in the ANFIS model. After that, the trained ANFIS model is used to forecast the numbers of cases for the next 5 days. The input and output variables are day number and number of infected cases, respectively.

In order to investigate the performance of the ANFIS model, the Root Mean Square Error mean (RMSE), normalized RMSE (NRMSE), Mean Absolute Percentage Error (MAPE), and coefficient of determination (R^2^) were used as follows [38,39]:(16)$$\begin{array}{c} \mathrm{RMSE}=\sqrt{\frac{1}{t}\overset{t}{\underset{t=0}{\sum }}{\left(y_{actual}-y_{estimated}\right)}^{2}},\;\mathrm{and}\\ \mathrm{NRMSE}\;\;\frac{RMSE}{y_{max}-y_{min}}, \end{array}$$
(17)$$\mathrm{MAPE}=\;\frac{y_{actual}-y_{estimated}}{y_{actual}}$$
(18)$$R^{2}=1-\frac{\sum_{t=0}^{t}{\left(y_{actual}-y_{estimated}\right)}^{2}}{\sum_{t=0}^{t}{\left(y_{actual}-y_{average}\right)}^{2}}.$$

## 3. Results

### 3.1. Infection Rate (β) Estimation

GA was applied to estimate the optimum infection rate between *0.2 ≤ β ≤ 0.4* by minimizing the cost function described in Equation (8). Figure 4 depicts the cost values for 1000 iterations. It is observed that the GA searching for the minimum cost value converges to the value of 1.098 × 10^−9^ at the iteration number 819, which indicates that there is no better cost value than 1.098 × 10^−9^ based on GA. The optimum *β* values obtained for the entire population size of 200 is shown in Figure 5. The *β* values are approximated by the normal distribution and, subsequently, the infection rate *β* is 0.228 ± 0.013. Based on Equation (5), the basic reproductive number is 2.28 ± 0.13 as *γ* = 0.1.

### 3.2. Epedimic Peak Prediction

Given that the major outbreak occurs after the second wave, it is assumed that the influence of the number of cases reported before the second wave is negligible in estimating the identification and infection rates. Besides, this assumption is considered due to the absence of the reported numbers related to cases that tested negative during the first wave. The epidemic peak is estimated when a maximum is attained within one year, such that *X(t_max_)* = max_0 < t < 365_
*X(t)*. Based on the current report, the *p* is around 0.084. Subsequently, Figure 6 shows a one-year behavior of *X(t)* for the determined infection rate *β* = 0.228 ± 0.013.

It is observed that the epidemic peak may occur between 170 (*β* = 0.241) and 200 (*β* = 0.215) with an average of 184 (*β* = 0.228). This indicate that, starting from 25 January, the predicted epidemic peak is on 26 July (*t* = 184), with deviation from 12 July (*t* = 170) to 11 August (*t* = 200). The COVID-19 pandemic will last until 15 December 2020 (*t* = 326), with the deviation ranging from 22 November 2020 (*t* = 303) to 12 January 2021 (*t* = 354).

Based on the entire period since the COVID-19 onset in Malaysia, the *p* value ranges from 0.01 to 0.084. Hence, we also estimate the epidemic peak at *p* = 0.01. Figure 7 shows the *X(t)* over one year for *β* = 0.228 ± 0.013. As seen, the predicted epidemic peak is 19 June (*t* = 147) and the uncertainty range is from 08 June (*t* = 136) to 02 July (*t* = 160). The COVID-19 pandemic will last until 29 September (*t* = 249) with the deviation ranging from 13 September (*t* = 233) to 19 October (*t* = 269). In contrast to the basic reproductive number ℛ_0_, it is clear that the epidemic peak and size are responsive to the identification rate *p*. Furthermore, a lower identification rate leads to a lower number of infected cases, such that the number of infected cases decreases from 2.582 × 10^5^ to 3.077 × 10^4^ at the epidemic peak with *p* = 0.01.

### 3.3. Epidemic Peak after Possible Interventions

In this subsection, the effect of possible interventions is investigated. In Malaysia, all universities, schools, and workshop places have been closed and most of the social events have been canceled from 17 to 26 March to eliminate the contact risk. However, the government has extended the closure to 14 April as the number of infected cases is still rising by an average of 170 cases per day. Thus, the current governmental effort seems to be limited to contain the COVID-19 up to now.

We assume that the governmental and social efforts can reduce the infection rate *β* = 0.228 by 25% of its value (*β_new_* = 0.17) during the period from 05 April (*t* = 72) to the desired day (*t* = *T* > 72), and we fix *p* to 0.084 in what follows. Firstly, it is assumed that the intervention is adopted for 2 months; that its, *T* is equal to 134 (72 + 62). In this situation, the epidemic peak *t_max_* is shifted 30 days later from 26 July to 26 August. It is clear that the epidemic size remains relatively unchanged. On the other hand, if the interventions are adopted for three months from 05 April to 04 July (*T* = 72 + 92 = 164), then the epidemic peak *t_max_* is moved back from 26 July to 09 September. It can be observed that the epidemic size is significantly reduced. Figure 8 shows the real-time prediction of infected cases that are identified between *t* = 0 and *t* = 365 for no intervention, two months intervention, and three months intervention.

We can also generalize the desired day for possible interventions over 72 < *T* < 365, as shown in Figure 9a. It is observed that the epidemic peak *t_max_* is linearly delayed as the intervention period increases from 72 ≤ *T* ≤ 263 and then fixed to *t_max_* for *T* > 263.

The figure also indicates that the interventions have a positive effect to delay the epidemic peak, which may give the government more time to contain the COVID-19 and flatten the curve. Figure 9b shows the relationship between the intervention period (*T*) and the number of infected cases at the epidemic peaks *X(t_max_)*. It is observed that the number of infected cases is monotonically declined and fixed as *T* increases. Interestingly, the change in the number of infected cases is rapidly increased for *T* > 72. This implies that an early intervention over a relatively small duration can be effective to reduce the epidemic size and flatten the curve.

### 3.4. Short-Term Forecasting

The ANFIS model was mainly used to forecast the infected cases for the next five days based on the historical data of 10 days. Firstly, the historical data is randomly split into training and testing datasets according to a 70%:30% ratio to make sure the model is not subjected to overfitting. Figure 10 shows the training and testing errors over the 300 epochs (iterations). Estimated (ANFIS output) and actual infected cases are depicted in Figure 11. Table 4 presents the RMSE, NRMSE, MAPE, and R^2^ obtained while training the ANFIS model using the training and testing datasets.

Secondly, the developed ANFIS model was then used to forecast the number of infected cases for the next five days. The results of the forecasted and actual number of infected cases are presented in Figure 12. The performance of the ANFIS model to forecast is as follows: the RMSE, NRMSE, MAPE, and R^2^ values are 96.8, 0.041, 2.45%, and 0.9964, respectively. These results indicate a very low RMSE, NRMSE, and MAPE, but a high R^2^.

## 4. Discussion

This study mainly aims to (1) estimate the infection rate using the GA algorithm; (2) predict the epidemic peak of COVID-19; and (3) forecast the number of infected cases for the upcoming five days based on historical data of the last ten days. First, the confirmed cases from 25 January to 05 April was used to find the coefficient parameters of the SEIR model. Subsequently, the GA was applied to find the infection rate value that minimizes the function of the SEIR model with Poisson noise. As a result, the infection rate is 0.228 ± 0.013. Based on Equation (5), the basic reproductive number ℛ_0_ is 2.28 ± 0.13. This value is relatively close to the estimated value by the World Health Organization (WHO), which ranges from 2 to 2.5 for COVID-19 [40]. In addition, this value is not so different from recent estimations: 2.24–3.58 [41], 2.0–3.1 [42], and 2.06–2.52 [43] for COVID-19. However, some studies reported higher ℛ_0_ values of 3.28, 2.90, and 3.11, as reported in [44,45,46], respectively. This bias in estimating the ℛ_0_ value is probably attributed to limited available data over a short period and also highly depends on the settings. Furthermore, the estimation of ℛ_0_ strongly relies on the estimation method and the validity of the assumptions for some coefficients. Thus, the availability of more data over a long period would provide a more accurate estimation and form a clearer trend.

Secondly, the SEIR model incorporating the mortality in the population due to COVID-19 was used to predict the epidemic peak of COVID-19 in this study. The epidemic peak in Malaysia could be reached late July 2020 and the uncertainty range is from 12 July to 11 August 2020. The results also indicated that the COVID-19 trend in Malaysia will not flatten too quickly. This indication might be consistent with the WHO’s statement [47] that COVID-19 is not a seasonal virus and thus will not disappear in the summer, such as the flu. It should be noticed that the epidemic estimation may be subjected to some variability, such that possible big change in social and natural situations would shorten the range of the peak estimation. Besides, the epidemic estimation relies on the mathematical modeling used to describe the epidemic. A complex model with more biological and epidemiological variables is more realistic. However, it requires more model parameters and coefficients to be estimated compared to a simpler one. Therefore, it is important to keep a balance between biological realism and eliminating the variability in the model prediction with a view to increase the reliability of the predictions.

The findings obtained for epidemic peak prediction are as follows: (1) the epidemic size is not affected by the identification rate, which ranges from 0.01 to 0.084 for the total population in Malaysia; (2) a near-future intervention has a great effect to postpone the epidemic peak that would give the government and healthcare providers more time to optimize the medical environment by training more staffs to deal with COVID-19; and (3) a longer period intervention should be taken into account to reduce the epidemic size. Although the Malaysian government has implemented the Movement Control Order (MCO) towards COVID-19 on 18 March throughout the country, the number of daily confirmed cases is still rising with an average of 170 cases for the last two weeks. Besides, more critical cases requiring intensive care units are being recorded. This trend is due to the following possible reasons:The number of people who had contact with COVID-19 patients is enormous, as reported in [48]. This could make the process of tracking and isolating more complex. Based on the information reported by Chinese medical doctors involved in Wuhan, the critical cases form 10% of the total number of infected people. The early diagnosis and treatment would reduce the flow of COVID-19 patients into the ICU unit [49].Poor experience in treating and managing cases with different levels of infection. For instance, severe cases should be kept under monitoring with intensive care, while mild cases without clear symptoms should be kept with less intensive care in the hospitals. However, patients under investigation should be placed in special isolation outside the hospitals. This kind of management would ease the treating process with the currently available equipment [50].The current MCO implemented in Malaysia is limited to aiding the awareness of the people to the danger of COVID-19. For the first 10 days of the MCO, 60% of the public has obeyed the MCO issued by the government [51]. Thus, more restrictions are needed to enforce the MCO. By increasing the public awareness, the infection rate will be reduced, which would result in decreasing the reproductive number and delaying the epidemic peak.

Lastly, this study provides short-term forecasting for the number of infected cases based on the ANFIS model. The results indicate a high forecast precision is achieved based on the ANFIS model. The ANFIS model achieved (1) an excellent coefficient of determination (R^2^ = 0.9964), which is very close to the perfect value of 1; (2) a low NRMSE value (NRMSE = 0.041), which is highly close to the perfect values of 0; and (3) a high MAPE value (MAPE = 2.45%), which is less than 10% [52]. The main motivation behind using the ANFIS model instead of parametric models (e.g., likelihood and Bayesian methods) is that ANFIS is able to achieve a high accuracy using only a few datasets and is easy to be deployed, such that the ANFIS model uses one input as day number, while parametric models require at least four inputs as well as estimation of the coefficients.

This study has some limitations. First, the SEIR model is used with a limited number of cases and COVID-19 is highly infectious; so, the current results of peak estimation are constrained to a limited period and may be changed after inputting a considerable number of infected cases. Secondly, the estimation is based on the available data from the WHO. A possible delay in confirming or reporting could result in an underestimation of ℛ_0_. Lastly, the ANFIS model is applicable for short-term forecasting, and so it cannot be used to predict the epidemic peak of COVID-19 as the ANFIS model does not consider the recovered and death rates.

## 5. Conclusions

As the main public concern in Malaysia is whether the COVID-19 spread will continue for the upcoming few months, we provide here information on predicting the epidemic peak using the SEIR model, estimating the infection rate using the GA algorithm, and short-time forecasting using the ANFIS model. The results related to the epidemic peak show that (1) the epidemic peak could be reached in the period ranging from 12 July to 11 August 2020, and last until the period ranging from 22 November 2020 to 12 January 2021; (2) the identification rate, which ranges from 0.01 to 0.084, does not affect the epidemic size for the total Malaysian population; (3) the influence of the identification rate on the basic reproductive number is negligible; and (4) a near-future intervention may decrease the infection rate, which would lead to a delay the epidemic peak. The results also show that the infection rate is 0.228 ± 0.013, while the basic reproductive number is 2.28 ± 0.13. Furthermore, a high forecasting accuracy is achieved, such that the NRMSE, MAPE, and R^2^ values are 0.041, 2.45%, and 0.9964, respectively.

## Figures and Tables

**Figure 1 ijerph-17-04076-f001:**
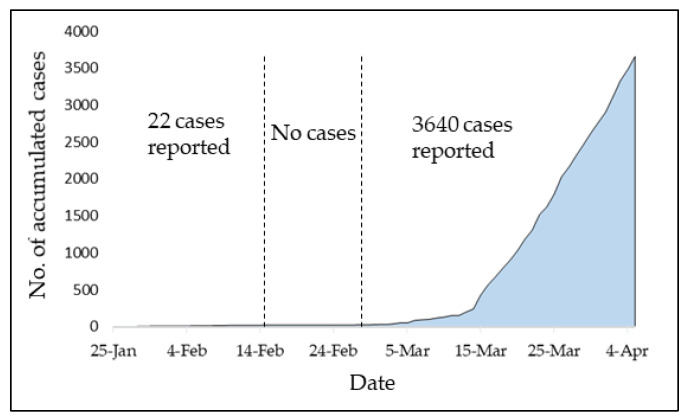
Growth in the total number of infected cases in Malaysia.

**Figure 2 ijerph-17-04076-f002:**
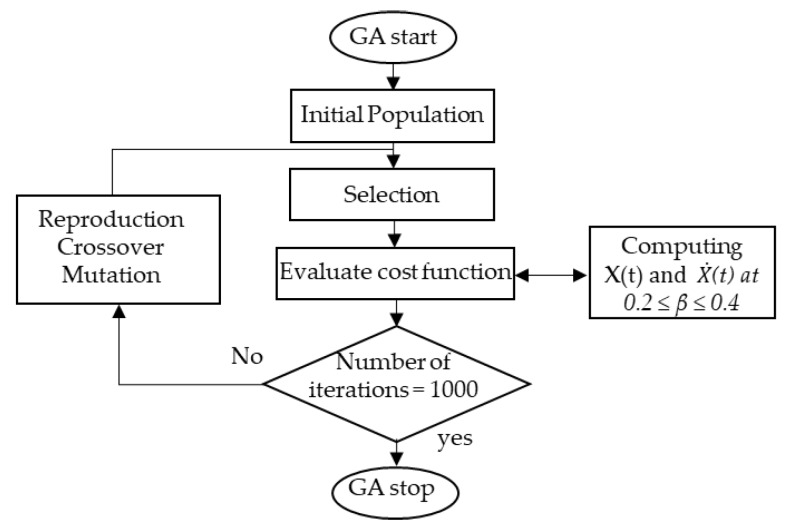
Genetic Algorithm (GA) flowchart for *β* estimation.

**Figure 3 ijerph-17-04076-f003:**
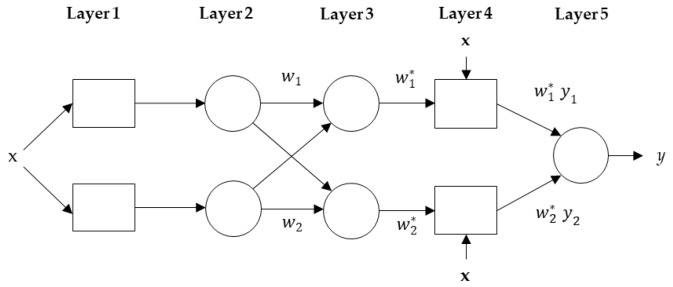
ANFIS structure.

**Figure 4 ijerph-17-04076-f004:**
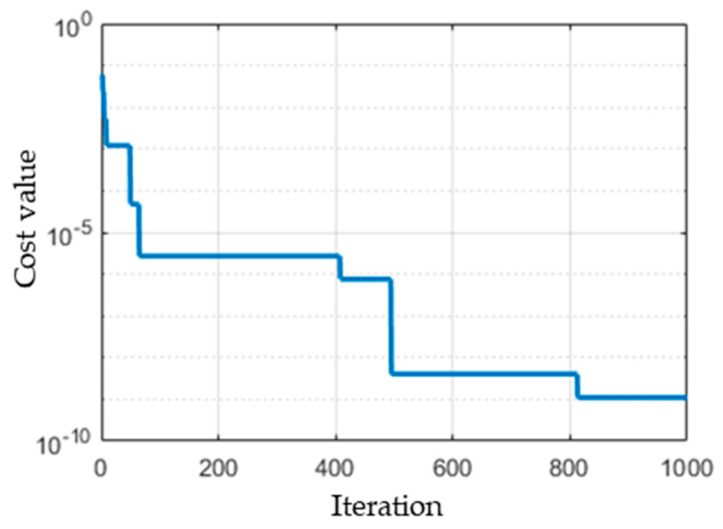
Cost values of 1000 iterations.

**Figure 5 ijerph-17-04076-f005:**
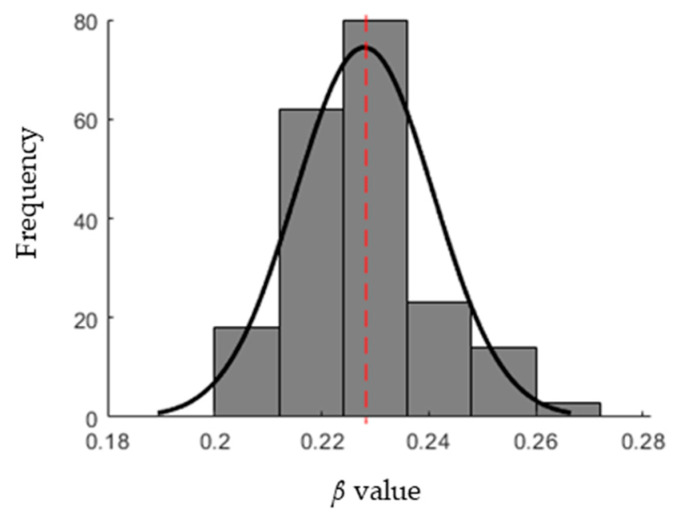
Normal distribution of optimum *β* values. The dotted line represents the mean value.

**Figure 6 ijerph-17-04076-f006:**
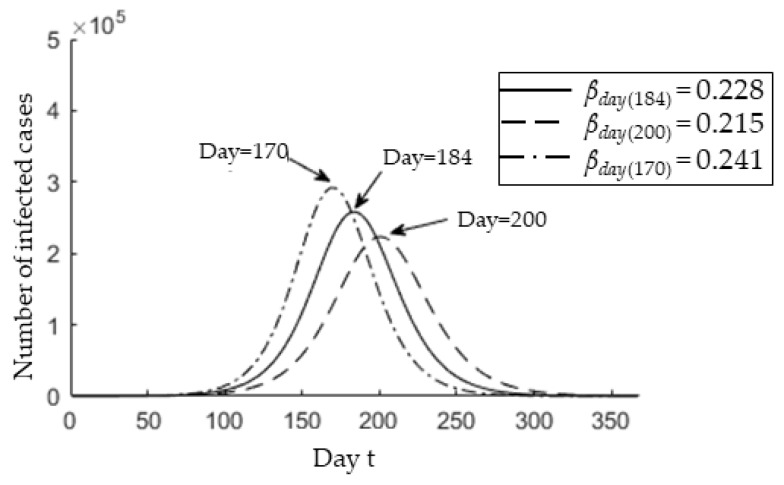
Real-time variation in the number of infected cases identified at time *t* for *p* = 0.084 and *β* = 0.215, *β* = 0.228, and *β* = 0.241. The text arrows represent the *t_max_* for each infection rate.

**Figure 7 ijerph-17-04076-f007:**
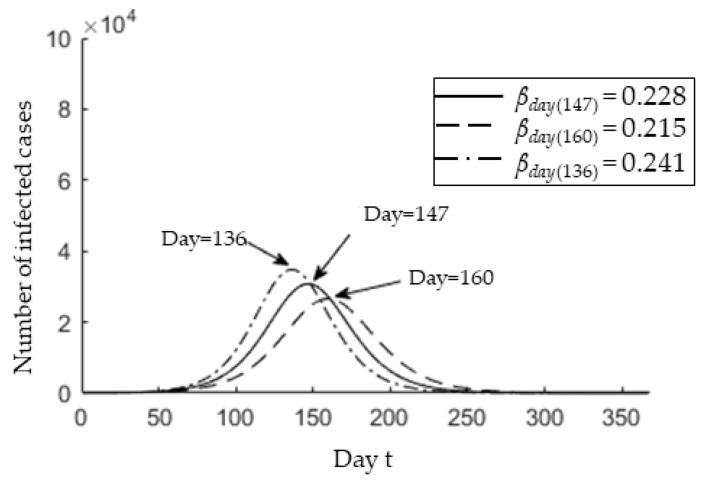
Real-time variation in the number of infected cases identified at time *t* for *p* = 0.01 and *β* = 0.215, *β* = 0.228, and *β* = 0.241. The text arrows represent the *t_max_* for each infection rate.

**Figure 8 ijerph-17-04076-f008:**
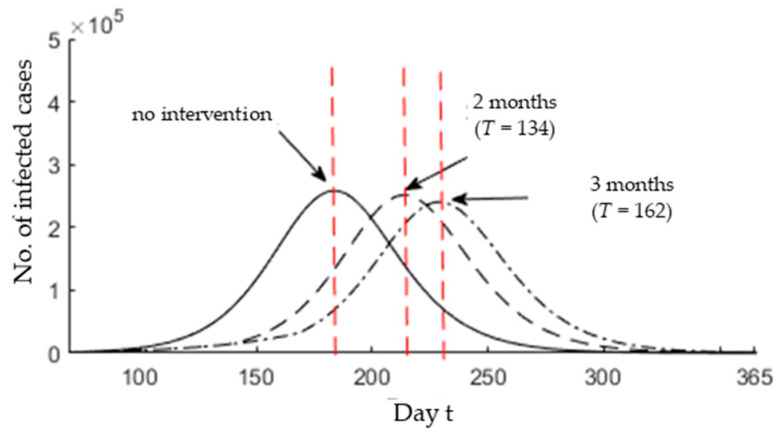
Real-time variation in the number of infected cases (0 ≤ *t* ≤ 365) for *p* = 0.084. The red dotted lines represent the epidemic peak.

**Figure 9 ijerph-17-04076-f009:**
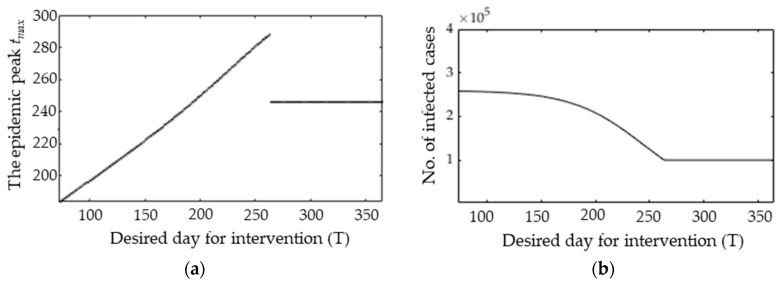
The relationship between the desired day for intervention T and (**a**) the epidemic peak *t_max_*; (**b**) the number of infected cases at epidemic peak *t_max_*.

**Figure 10 ijerph-17-04076-f010:**
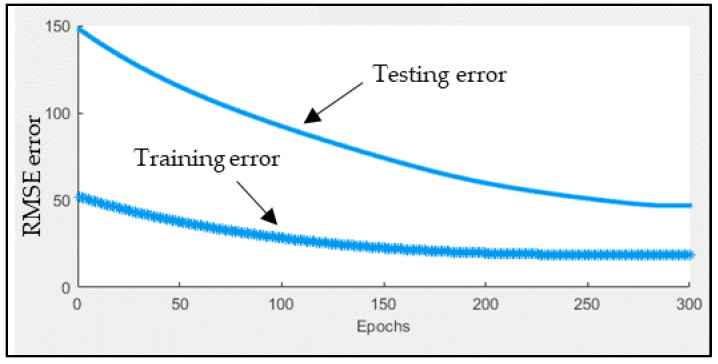
The upper and lower curves represent the training and testing errors, respectively.

**Figure 11 ijerph-17-04076-f011:**
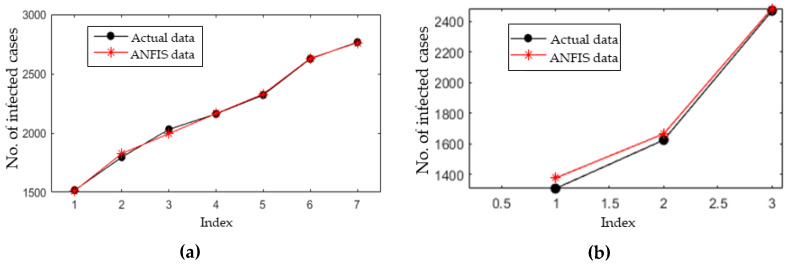
Estimated and actual infected cases using the (**a**) training dataset and (**b**) testing dataset.

**Figure 12 ijerph-17-04076-f012:**
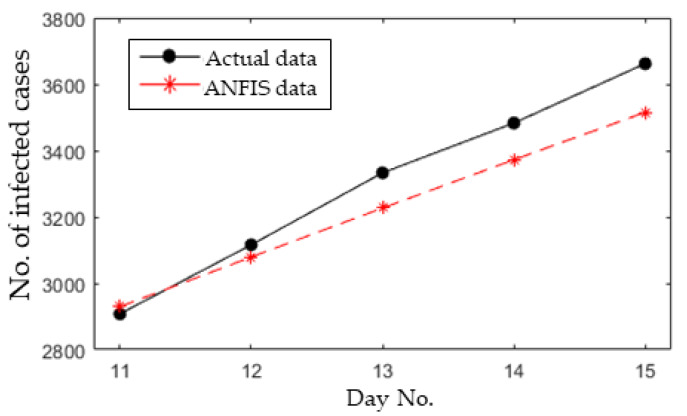
Forecasting results for the next five days.

**Table 1 ijerph-17-04076-t001:** Coefficient values for the Susceptible–Exposed–Infectious–Recovered (SEIR) model.

Coefficient	Description	Value
*α*	Onset rate	0.2
*γ*	Removal rate	0.1
*M*	Mortality rate	0.016
*N*	Malaysia population	32.6 × 10^6^
*p*	Identification rate	0.084

**Table 2 ijerph-17-04076-t002:** GA parameters.

Parameter	Value	Parameter	Value
Population size	200	Mutation rate	0.02
Number of iterations	1000	Mutation percentage	0.9
Crossover percentage	0.95		

**Table 3 ijerph-17-04076-t003:** Adaptive Neuro-Fuzzy Inference System (ANFIS) parameters.

Parameter	Method/Value	Parameter	Method/Value
Fuzzy structure	Sugeno-type	No. of epochs	300
Rules clustering	Grid partition	Input	Day number
MF type	Gaussian	Output	Infected cases
Optimization method	Hybrid	Output MF	constant

**Table 4 ijerph-17-04076-t004:** Performance of the ANFIS model.

Parameter	Training Data	Testing Dataset
RMSE	18.53	46.87
NRMSE	0.012	0.032
MAPE	1.31%	2.79%
R^2^	0.9973	0.9998

## Data Availability

The data and MATLAB^®^ codes used to generate the results are available from the corresponding author upon request.

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
