# Peer review of "Prediction of Epidemic Peak and Infected Cases for COVID-19 Disease in Malaysia, 2020"

_ijerph, 2020, doi:10.3390/ijerph17114076_

Round 1

Reviewer 1 Report

Summary

The authors predict the epidemic peak for COVID-19 in Malaysia. They use genetic algorithm to estimate the infection rate. They also predict the infected number of cases in short term using an adaptive neuro-fuzzy inference system model. They also conducted sensitivity analysis on the epidemic peak based on the possible interventions. Their results show high prediction accuracy. The manuscript is written well and easy to follow. Overall, the paper contains important information about the novel Coronavirus and what it means for Malaysia. I thank the authors for their work. Below are some suggestions that would improve the paper. I put the suggestions by sections below.

Abstract: Mentioning the impact in numbers in the abstract would be nice as it is already expected to have a great impact.

  1. Introduction:

The COVID-19 spread depends highly on the characteristics of the country. The paper focuses on how COVID-19 has spread in Malaysia. I am not much familiar with the general characteristics of Malaysia. Authors mention about the religious gathering- so the are there any other religious events that might have impacted the spread? Or like any imported cases have occurred? So for example, Hajj was delayed starting April 1st.

In addition, what is the county’s general attitude towards social distancing? For example, we know that for certain countries social distancing is much harder than others.

What is the availability of resources, especially the testing kits?

  1. Methods

Can you elaborate more on the p: identification rate? This is basically among the number of cases tested what portion of them are positive. This does not include the number of available kits and assumes that everybody who might have the symptoms can be tested. Is this a reasonable assumption? Again, I am not knowledgeable about the country’s testing kits and their policy on testing whether they are doing selective testing or not but please comment/clarify on this.

It is not that important but the paper from John Hopkins state 11.5 days of infectious period.

  1. Results

3.2 Epidemic Peak prediction

Why does the epidemic peak prediction assume “within one year”? Is this something related to Malaysia? Discussions in some other counties are talking about potential school openings in 2021.

3.3 Epidemic Peak After Possible Interventions

What is the relationship between delay in the disease for interventions and the response from the public? Section 4 mentions that 60% only obeyed the MCO so far. Does this assume the same behavior is going to continue?

  1. Discussion:

I appreciate this part a lot as it talks about Malaysia specific details.

I am a bit curious and confused about underlying reason for the following statement:

“the epidemic size is not affected by the identification rate which ranges from 0.01 to 0.084 for the total population in Malaysia”

Particularly, how much of this statement is dependent on the test kit availability?

Also, in section 3, you have “In contrast to the basic reproductive number, it is clear that the epidemic peak and size are responsive to the identification rate p.” Well, could you please explain the reason behind the difference in these two sentences?

In the discussion section, I would assume that in addition to the limited data over a short period of time, the bias also highly depends on the setting.

The possible reason #3 states that that 60% of the population has obeyed the MCO. What would an increase in that number change in the results?

  1. Conclusion: Comments are similar to the discussion.

Author Response

We would like to thank you for your time and efforts to improve the article. We respond to the suggestions as in the attached file. 

Reviewer 2 Report

This manuscript models the time course expected from the  COVID19 epidemic in  Malaysia.. It is a paper of considerable interest both to Malaysians and to the rest of the world. However, it is very technical with a lot of mathematics that should be accompanied by actual numbers and proportions of the community that the results relate. Similarly, their should be sensitivity analysis with numerical examples of what changes in estimated parameters used in the equations would relate to in terms of numbers of subjects at, exposed etc.

The issue of second wave  responses needs to be address- both in terms of the residual virus in the community and the risk of community spread may look like. Are there calculations applicable to other countifies? How long do the authors predict the panepidemic to last?

Author Response

We would like to thank you for your time and efforts to evaluate the article. our response is attached below.

Author Response

We would like to thank you for your time and efforts to review the article. The response is attached below. 

Reviewer 4 Report

In the present work, the authors provide a prediction of the epidemic peak and the number of infected cases of COVID-19 in Malaysia. Overall, the paper is well written, and the topic tackled in this work is of utmost importance. Indeed, the prediction of the behavior of this disease is the main topic of discussion on these days and, in that sense, it is a hot topic'.

The work is carried out using a classical SEIR model, which is a system of ordinary differential equations describing the dynamic of the amount of susceptible, exposed, infected and recovered. From the mathematical point of view, there is no novelty in this work: it is just an application of a well-known model available in the literature. The authors basically substitute the values of the parameters (known from the empirical information available) and the initial conditions.

On the other hand, it is important to point out that the mathematical model (1) has been already used in the literature to study the evolution of COVID-19 in Japan and Italy (see references 9 and 10).

In view of all of these comments (the lack of novelty and interesting results), I recommend the paper for REJECTION.

Author Response

We would like to thank you for reviewing our article. The response to your comments is attached below. 

Round 2

Reviewer 3 Report

The authors have responded satisfactorily to my comments and therefore I think the paper can be published in its present form

Author Response

Thank you so much for your time and efforts that improve the article. 

best regards